# Corneal confocal microscopy identifies greater corneal nerve damage in patients with a recurrent compared to first ischemic stroke

**Adnan Khan**[1], **Naveed Akhtar**[2], **Saadat Kamran**[2], **Hamad Almuhannadi**[1], **Georgios Ponirakis**[1], **Ioannis N. Petropoulos**[1], **Blessy Babu**[2], **Namitha R. Jose**[2], **Rumissa G. Ibrahim**[2], **Hoda Gad**[1], **Paula Bourke**[2], **Maher Saqqur**[2,3], **Ashfaq Shuaib**[3], **Rayaz A. Malik**[1]*

1 Department of Medicine, Weill Cornell Medicine-Qatar, Doha, Qatar, 2 Department of Neurology and Institute of Neurosciences, Hamad Medical Corporation, Doha, Qatar, 3 Department of Neurology, Stroke Program, University of Alberta, Alberta, Canada

* ram2045@qatar-med.cornell.edu

**Data Availability Statement:** The data used for statistical analysis in this study is available (https://figshare.com/s/ea4479b2063a26113cf0).

## Abstract

### Objectives

Corneal nerve damage may be a surrogate marker for the risk of ischemic stroke. This study was undertaken to determine if there is greater corneal nerve damage in patients with recurrent ischemic stroke.

### Methods

Corneal confocal microscopy (CCM) was used to quantify corneal nerve fiber density (CNFD), corneal nerve branch density (CNBD), corneal nerve fiber length (CNFL) and corneal nerve fiber tortuosity (CNFT) in 31 patients with recurrent ischemic stroke, 165 patients with a first acute ischemic stroke and 23 healthy control subjects.

### Results

Triglycerides (P = 0.004, P = 0.017), systolic BP (P = 0.000, P = 0.000), diastolic BP (P = 0.000, P = 0.000) and HbA$_{1c}$ (P = 0.000, P = 0.000) were significantly higher in patients with first and recurrent stroke compared to controls. There was no difference in age, BMI, HbA$_{1c}$, total cholesterol, triglycerides, LDL, HDL, systolic and diastolic BP between patients with a first and recurrent ischemic stroke. However, CNFD was significantly lower (24.98±7.31 vs 29.07±7.58 vs 37.91±7.13, P<0.05) and CNFT was significantly higher (0.085±0.042 vs 0.064±0.037 vs 0.039±0.022, P<0.05) in patients with recurrent stroke compared to first stroke and healthy controls. CNBD (42.21±24.65 vs 50.46±27.68 vs 87.24±45.85, P<0.001) and CNFL (15.66±5.70, P<0.001 vs 17.38±5.06, P = 0.003) were equally reduced in patients with first and recurrent stroke compared to controls (22.72±5.14).

**Funding:** Supported by Qatar National Research Fund Grant BMRP20038654 to Dr Rayaz A. Malik. The funders had no role in study design, data collection and analysis, decision to publish, or publish, or preparation of the manuscript.

**Competing interests:** The authors have declared that no competing interests exist.

## Conclusions

Corneal confocal microscopy identified greater corneal nerve fibre loss in patients with recurrent stroke compared to patients with first stroke, despite comparable risk factors. Longitudinal studies are required to determine the prognostic utility of corneal nerve fiber loss in identifying patients at risk of recurrent ischemic stroke.

## Introduction

Recurrent stroke occurs in 11.1% of stroke patients within one year of the initial stroke [1], and is associated with greater disability and mortality [2]. A recent study from China has shown that the incidence of recurrent stroke has increased 3-fold between 1992 and 2012 [3]. Age [4], dyslipidemia [5], smoking [6], diabetes, hypertension, homocysteine levels, atrial fibrillation [1], metabolic syndrome [7] and other risk factors [8–10] are associated with recurrent stroke. Indeed, a recent study has shown that hypertension, prior symptomatic stroke and chronic infarcts on MRI were independently associated with recurrent stroke and this also doubled the all-cause mortality [11]. However, an artificial neural network model utilizing 19 independent variables generated only a moderate accuracy of 75% for predicting stroke recurrence at 1-year [12].

Brain imaging reveals that the presence of multiple white matter hyperintensities [13–15], silent lacunar infarcts and isolated cortical lesions are associated with recurrent stroke and the presence of white matter hyperintensities, micro-bleeds [16] and silent new ischemic lesions [17, 18] predict the risk of stroke [19]. Furthermore, the 5-year recurrent stroke risk in the presence of severe white matter changes is comparable to the presence of atrial fibrillation and hypertension [20].

Corneal confocal microscopy (CCM) is a noninvasive ophthalmic imaging technique for rapid, high-resolution imaging of corneal nerves. This technique has identified axonal loss in diabetes [21–23], impaired glucose tolerance [24], other peripheral neuropathies [25, 26], Parkinson's disease [27], amyotrophic lateral sclerosis [28], multiple sclerosis [29] and dementia [30]. More recently we have shown a significant loss of corneal nerves in patients with TIA [31] and acute ischemic stroke [32–34], which was associated with elevated triglycerides and $HbA_{1c}$. Vascular risk factors including dysglycemia and dyslipidemia [35] and hypertension [36] are associated with corneal nerve loss and an improvement in blood pressure, lipids, $HbA_{1c}$ [37] and glucose tolerance [38] are associated with an improvement in corneal nerve morphology.

Given that there are shared risk factors for stroke and corneal nerve loss, we hypothesized that patients with recurrent ischemic stroke will demonstrate greater corneal nerve abnormality compared to those with first ischemic stroke, reflecting the greater overall exposure to the risk factors for stroke.

## Materials and methods

Thirty-one patients with a recurrent acute ischemic stroke, 165 patients with a 1st acute ischemic stroke and 23 age-matched healthy control participants were studied. The diagnosis of stroke was confirmed clinically and radiologically using AHA criteria [39]. Exclusion criteria included patients with intracerebral hemorrhage, a known history of ocular trauma or surgery, high refractive error, glaucoma, dry eye and corneal dystrophy [40]. Demographic (age,

gender, ethnicity) and clinical (blood pressure, $HbA_{1c}$, lipid profile) data were obtained from patients' health records. All patients underwent assessment of the National Institutes of Health Stroke Scale (NIHSS) at presentation. This study adhered to the tenets of the declaration of Helsinki and was approved by the Institutional Review Board of Weill Cornell Medicine (15–00021) and Hamad General Hospital (15304/15). Informed, written consent was obtained from all patients/guardians before participation in the study.

## Corneal confocal microscopy

All patients underwent CCM (Heidelberg Retinal Tomograph III Rostock Cornea Module; Heidelberg Engineering GmbH, Heidelberg, Germany). CCM uses a 670 nm wavelength helium neon diode laser, which is a class I laser and therefore does not pose any ocular safety hazard. A ×63 objective lens with a numeric aperture of 0.9 and a working distance, relative to the applanating cap (TomoCap; Heidelberg Engineering GmbH) of 0.0 to 3.0 mm, is used. The size of each 2-dimensional image produced is 384×384 pixels with a 15˚×15˚ field of view and 10 μm/pixel transverse optical resolutions. To perform the CCM examination, local anesthetic (0.4% benoxinate hydrochloride; Chauvin Pharmaceuticals, Chefaro, United Kingdom) was used to anesthetize both eyes, and Viscotears (Carbomer 980, 0.2%, Novartis, United Kingdom) was used as the coupling agent between the cornea and the cap. Patients were asked to fixate on an outer fixation light throughout the CCM scan and a CCD camera was used to correctly position the cap onto the cornea [41]. The examination took approximately 10 minutes for both eyes. The examiners captured images of the central sub-basal nerve plexus using the section mode. On the basis of depth, contrast, focus, and position, 6 images per patient were selected [42]. All CCM images were manually analyzed using validated, purpose-written software. Corneal nerve fiber density (CNFD), corneal nerve branch density (CNBD), corneal nerve fiber length (CNFL) and corneal nerve fiber tortuosity (CNFT) were analyzed using CCMetrics (M. A. Dabbah, ISBE, University of Manchester, Manchester, United Kingdom) [21].

## Statistical analysis

All statistical analyses were performed using IBM SPSS Statistics software Version 25. Normality of the data was assessed using the Shapiro-Wilk test and by visual inspection of the histogram and a normal Q-Q plot. Data are expressed as mean ± standard deviation (SD). Each group was compared using ANOVA (for normally distributed variables) with Bonferroni as post hoc test and the non-parametric Kruskal-Wallis test (for non-normally distributed variables). To investigate the association between risk factors for corneal nerve parameters, Pearson and Spearman correlation were performed as appropriate. Multiple linear regression analysis was conducted to evaluate the independent association between corneal nerve loss and their covariates. The data used for statistical analysis in this study is available (https://figshare.com/s/ea4479b2063a26113cf0).

## Results

### Clinical and metabolic characteristics

The clinical and metabolic characteristics of the cohorts of participants studied are summarized in Table 1.

Thirty-one patients with a recurrent ischemic stroke were compared with 165 patients with a 1st ischemic stroke and 23 age-matched healthy controls. There was no significant difference in the percentage of patients with a 1st stroke compared to recurrent (2nd) stroke in relation to

**Table 1. Clinical, demographic, metabolic and CCM measures in study participants.**

| Characteristics | Controls | 1st Stroke | Recurrent Stroke |
|---|---|---|---|
| Number of Participants | 23 | 165 | 31 |
| Age (years) | 52.43 ± 14.59 | 49.34 ± 9.49 | 50.49 ± 9.47 |
| BMI (kg/m$^2$) | 26.17 ± 1.60 | 27.68 ± 4.63 | 28.32 ± 5.39 |
| NIHSS Score | N/A | 4.66 ± 4.64 | 4.39 ± 3.11 |
| Triglycerides (mmol/l)** | 1.08 ± 0.59 | 1.81 ± 1.17‡ | 2.06 ± 1.54‡ |
| Total Cholesterol * (mmol/l) | 4.12 ± 1.66 | 5.06 ± 1.20‡ | 4.90 ± 1.24 |
| LDL (mmol/l) | 2.86 ± 1.01 | 3.29 ± 1.09 | 2.99 ± 0.95 |
| HDL (mmol/l) * | 1.19 ± 0.30 | 0.95 ± 0.25 | 0.94 ± 0.25 |
| BP Systolic (mmHg)*** | 125.40 ± 13.40 | 158.38 ± 28.67‡ | 160.39 ± 36.34‡ |
| BP Diastolic (mmHg)*** | 75.07 ± 8.42 | 93.19 ± 16.35‡ | 93.84 ± 18.14‡ |
| HbA$_{1c}$ (%) | 5.66 ± 0.32 | 6.83 ± 2.19‡ | 7.19 ± 2.79‡ |
| CNFD (no./mm$^2$)*** | 37.91 ± 7.13 | 29.07 ± 7.58‡ | 24.98 ± 7.31‡† |
| CNBD (no./mm$^2$)*** | 87.24 ± 45.85 | 50.46 ± 27.68‡ | 42.21 ± 24.65‡ |
| CNFL (mm/mm$^2$) *** | 22.72 ± 5.14 | 17.38 ± 5.06‡ | 15.66 ± 5.70‡ |
| CNFT (TC)*** | 0.039 ± 0.022 | 0.064 ± 0.037‡ | 0.085 ± 0.042‡† |

Results are expressed as mean ± SD. Statistically significant differences between groups using ANOVA

* P<0.05

** P<0.01

*** P<0.001.

‡ Post hoc results differ significantly from control group (P<0.05).

† Post hoc results differ significantly from 1st stroke group (P<0.05).

the use of statins (95% vs 87%), ACE-inhibitors (52% vs 58%), Angiotensin II receptor blockers (6% vs 13%), Beta blockers (10% vs 29%), calcium channels blockers (16% vs 26%), aspirin (83% vs 77%) and clopidogrel (58% vs 45%) (Table 2).

## Clinical and metabolic variables in patients with a 1st stroke, recurrent stroke and healthy controls

Systolic BP (P = 0.000), diastolic BP (P = 0.000, P = 0.000), HbA$_{1c}$ (P = 0.000, P = 0.000), total cholesterol (P = 0.035, P = 0.196) and triglycerides (P = 0.004, P = 0.017) were significantly higher in the patients with a 1st stroke and recurrent stroke (except cholesterol) compared to healthy controls (Table 1).

**Table 2. Percentage of patients on different medications.**

| Medications | 1st Stroke | 2nd Stroke |
|---|---|---|
| Statins (%) | 153/165 (95%) | 27/31 (87%) |
| ACE Inhibitors (%) | 85/165 (52%) | 18/31 (58%) |
| ARB's (%) | 09/165 (6%) | 4/31 (13%) |
| Beta blockers (%) | 17/165 (10%) | 09/31 (29%) |
| Calcium channel blockers (%) | 26/165 (16%) | 5/31 (26%) |
| Aspirin (%) | 137/165 (83%) | 24/31 (77%) |
| Clopidogrel (%) | 96/165 (58%) | 14/31 (45%) |

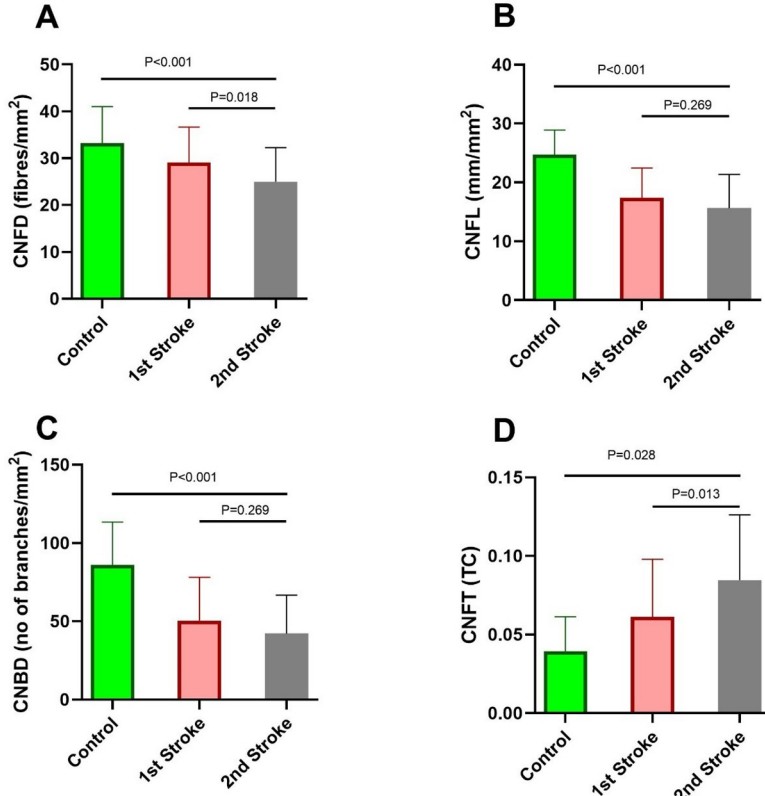

**Fig 1. Corneal nerve fiber parameters in patients with 1st ischemic stroke, recurrent (2nd) ischemic stroke and control subjects. (A)** CNFD: Corneal nerve fiber density; **(B)** CNFL: Corneal nerve fiber length; **(C)** CNBD: Corneal nerve branch density; **(D)** CNFT: Corneal nerve fiber tortuosity. Data are expressed as mean ± SD.

## Clinical and metabolic variables in recurrent v 1st stroke

There was no significant difference in age, BMI, HbA$_{1c}$, total cholesterol, triglycerides, LDL, HDL, systolic and diastolic BP between patients with a first and recurrent ischemic stroke (Table 1).

## CCM in patients with a recurrent stroke, 1st stroke and healthy controls

CNFD (P<0.001, P<0.001), CNFL (P<0.001, P<0.001) and CNBD (P = 0.003, P<0.001) were significantly lower, and CNFT (P = 0.028, P<0.001) was significantly higher in patients with 1st and recurrent stroke compared to healthy controls (Table 1, Fig 1).

## CCM in recurrent v 1st stroke

CNFD (P = 0.018) was significantly lower and CNFT (P = 0.013) was significantly higher in patients with recurrent stroke compared to 1st stroke. There was no significant difference in CNFL (P = 0.269) or CNBD (P = 0.269) between patients with recurrent compared to 1st stroke (Table 1, Fig 1).

## Multiple linear regression

There were independent associations between corneal nerve and metabolic parameters in patients with stroke (Table 3). CNFD was significantly associated with age (β = −0.204,

**Table 3. Independent risk factors for altered corneal nerves in patients with acute ischemic stroke.**

|  | B | 95% CI | SE | *P-Value* |
|---|---|---|---|---|
| **CNFD (fibers/mm$^2$)** | **50.689** | **(39.601–61.776)** | **5.611** | **<0.001** |
| Age (years) | -0.204 | (-0.320 – -0.088) | 0.059 | <0.000 |
| BMI (kg/m$^2$) | -0.525 | (-0.775 – -0.276) | 0.126 | 0.001 |
| BP Diastolic (mmHg) | 0.082 | (0.017–0.147) | 0.033 | 0.014 |
| Stroke | -2.180 | (-3.605 – -0.756) | 0.721 | 0.003 |
| **CNFL (mm/mm$^2$)** | **31.395** | **(24.830–37.961)** | **3.323** | **0.000** |
| Age (years) | -0.105 | (-0.189 – -0.021) | 0.043 | 0.015 |
| BMI (kg/m$^2$) | -0.333 | (-0.514 – -0.151) | 0.092 | 0.000 |
| **CNFT (TC)** | 0.036 | (0.015–0.057) | 0.011 | 0.001 |
| NIHSS at Admission | 0.001 | (0.000–0.003) | 0.001 | 0.049 |
| Stroke | 0.010 | (0.001–0.018) | 0.004 | 0.022 |

P<0.001), BMI (β = –0.525, P = 0.001) and diastolic BP (β = 0.082, P = 0.014). CNFL was significantly associated with age (β = –0.105, P = 0.015) and BMI (β = –0.333, P = 0.000). CNFT was significantly associated with NIHSS (β = 0.001, P = 0.049). CNBD was skewed, therefore it was not included in the regression analysis.

## Discussion

There is a need to identify risk factors or surrogate markers for stroke recurrence, such that high risk individuals can be targeted for more aggressive risk factor reduction. This is the first study to show greater corneal nerve loss in patients with recurrent ischemic stroke compared to a 1$^{st}$ ischemic stroke. This extends our previous findings demonstrating corneal nerve loss in patients with TIA [31] and acute ischemic stroke [32, 33].

Individual vascular risk factors such as diabetes, hypertension, smoking, dyslipidemia and metabolic syndrome are associated with the risk of a first and recurrent ischemic stroke [5–7, 13, 43]. A recent study has shown that a greater increase in carotid intima media thickness (IMT) was associated with an increased incidence of major adverse cerebral and coronary events [44]. Similarly, in the J-STARS (Japan Statin Treatment Against Recurrent Stroke) study, patients with the greatest baseline IMT were at the highest risk of recurrent stroke, which was partially ameliorated by treatment with pravastatin [45]. Intervention with dual as opposed to single antiplatelet therapy reduces the risk of recurrent stroke, but it is also associated with an increased risk of adverse events [46]. It is therefore important to identify those patients who may benefit the most from more aggressive control of risk factors. Interestingly, a recent longitudinal study of patients with a myocardial infarction from Stockholm showed that whilst albuminuria was associated with an increased risk of recurrent myocardial infarction there was no association with ischemic stroke [47].

MRI studies have shown that structural alterations including white matter hyperintensities, lacunes and microbleeds are associated with an increased risk of recurrent ischemic stroke [9, 13, 15]. We have previously shown a loss of corneal nerves in subjects with a major ischemic stroke compared to controls and an association between corneal nerve loss with HbA1c and triglycerides [32, 33]. In the present study, despite age, BMI, HbA$_{1c}$, lipids, BP and use of medications to treat blood pressure and lipids being comparable between those with recurrent stroke and 1$^{st}$ stroke, there was greater corneal nerve damage in patients with recurrent compared to 1$^{st}$ stroke. This suggests that corneal nerve loss may reflect the cumulative effect of known vascular risk factors and unknown risk factors for stroke and act as a surrogate marker for the risk of stroke and recurrent stroke.

This study is limited by the modest number of patients studied with recurrent ischemic stroke. We were also not able to include patients with severe stroke as CCM could not be performed in these individuals, due to their inability to cooperate during the CCM procedure. Whilst this may limit the utility of CCM across the spectrum of severity of stroke, it may also have biased the outcomes as the results may have been even more pronounced in those with more severe stroke. Nevertheless, we show greater corneal nerve abnormalities in patients with recurrent compared to a 1st acute ischemic stroke. Larger, longitudinal studies assessing corneal nerve fibre morphology in those at higher risk of stroke, perhaps with TIA and in relation to therapies to reduce risk factors for stroke are warranted to establish the clinical utility of corneal confocal microscopy in ischemic stroke.

## Author Contributions

**Conceptualization:** Adnan Khan.

**Data curation:** Adnan Khan, Georgios Ponirakis, Blessy Babu, Namitha R. Jose, Rumissa G. Ibrahim, Paula Bourke.

**Formal analysis:** Adnan Khan.

**Funding acquisition:** Rayaz A. Malik.

**Investigation:** Adnan Khan, Naveed Akhtar, Saadat Kamran, Hamad Almuhannadi, Ioannis N. Petropoulos, Hoda Gad, Maher Saqqur.

**Methodology:** Adnan Khan, Ashfaq Shuaib, Rayaz A. Malik.

**Project administration:** Naveed Akhtar, Ashfaq Shuaib, Rayaz A. Malik.

**Resources:** Naveed Akhtar, Ashfaq Shuaib, Rayaz A. Malik.

**Software:** Adnan Khan.

**Supervision:** Ashfaq Shuaib, Rayaz A. Malik.

**Validation:** Adnan Khan, Ashfaq Shuaib, Rayaz A. Malik.

**Visualization:** Adnan Khan.

**Writing – original draft:** Adnan Khan.

**Writing – review & editing:** Adnan Khan, Ashfaq Shuaib, Rayaz A. Malik.

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
