## [Decision Letter · Decision Letter 0]

19 Mar 2020

PONE-D-20-03573

Corneal Confocal Microscopy Identifies Greater Corneal Nerve Damage in Patients with a Recurrent Compared to First Ischemic Stroke

PLOS ONE

Dear Dr. Malik,

Thank you for submitting your manuscript to PLOS ONE. After careful consideration, we feel that it has merit but does not fully meet PLOS ONE’s publication criteria as it currently stands. Therefore, we invite you to submit a revised version of the manuscript that addresses the points raised during the review process.

Two Reviewers well assessed this manuscript.  However, several major revisions are needed in the present form.  See the Reviewers’ comments and respond them appropriately.

We would appreciate receiving your revised manuscript by May 03 2020 11:59PM. To enhance the reproducibility of your results, we recommend that if applicable you deposit your laboratory protocols in protocols.io, where a protocol can be assigned its own identifier (DOI) such that it can be cited independently in the future. For instructions see: http://journals.plos.org/plosone/s/submission-guidelines#loc-laboratory-protocols

We look forward to receiving your revised manuscript.

Kind regards,

Masaki Mogi

Academic Editor

PLOS ONE

Journal Requirements:

"Funding

Supported by Qatar National Research Fund Grant BMRP20038654 to Dr Rayaz A. Malik. The

funders had no role in study design, data collection and analysis, decision to publish, or publish,

or preparation of the manuscript.".

i) We note that you have provided funding information that is not currently declared in your Funding Statement. However, funding information should not appear in the Acknowledgments section or other areas of your manuscript. We will only publish funding information present in the Funding Statement section of the online submission form.

ii) Please remove any funding-related text from the manuscript and let us know how you would like to update your Funding Statement. Currently, your Funding Statement reads as follows:

 "No".

"No". 

Reviewers' comments:

Reviewer's Responses to Questions

**Comments to the Author**

1. Is the manuscript technically sound, and do the data support the conclusions?

Reviewer #1: Yes

Reviewer #2: Partly

2. Has the statistical analysis been performed appropriately and rigorously? 

Reviewer #1: Yes

Reviewer #2: I Don't Know

3. Have the authors made all data underlying the findings in their manuscript fully available?

Reviewer #1: Yes

Reviewer #2: Yes

4. Is the manuscript presented in an intelligible fashion and written in standard English?

Reviewer #1: Yes

Reviewer #2: Yes

5. Review Comments to the Author

Reviewer #1: This paper provides new data suggesting that corneal confocal microscopy demonstrates greater corneal nerve damage in patients with recurrent stroke compared to the patients with first stroke and healthy control subjects. The study protocol is correct, the used technology is innovative and the paper is well written.

Below are my minor recommendations:

- Introduction (Page 3, Line 4): The word “smoking” has been repeated twice in the same sentence.

- Introduction (Page 3, Paragraph 2, Line 1): Please edit as “… multiple white matter …”

- Materials and Methods (Page 4, Line 5): Please use commas rather than semicolons.

- Materials and Methods (Page 5, Lines 2,3): “The size of each 2-dimensional image produced is 384x384 μm…” Please edit as “384x384 pixels” or “400x400 µm”.

Reviewer #2: The authors describe a possible prognostic utility of corneal nerve fiber loss in patients with recurrent stroke.

The paper is overall well-written, the methods and results are presented in a concise and easy-to-follow way. The findings are of interest, although the larger studies are warranted to establish the clinical utility of corneal confocal microscopy in ischemic stroke, as the authors also pointed out in the discussion.

There are couple of points that are missing or should be described more clearly:

1. Materials and Methods:

How was the healthy control subject defined. Were there any inclusion/ exclusion criteria except no history of stroke? Given the age of the participants, I would assume that they were not all completely without any health problem or medication? Please, specify this.

2. Discussion

The corneal nerve fiber loss has largely been studied in diabetic patients. It was hypothesised in these patients that the corneal nerve fiber damage may reflect the damage of peripheral nerve system mainly, and the retinal nerve fiber loss (assessed e.g. by optical coherence tomography) may reflect the CNS damage better. Please, discuss a possible etiopathogenetic mechanism that advocates the notion that corneal nerves status is closely tied to the risk of recurrent stroke more clearly.

6. PLOS authors have the option to publish the peer review history of their article (what does this mean?). If published, this will include your full peer review and any attached files.

Reviewer #1: No

Reviewer #2: No

---

## [Author Response · Author response to Decision Letter 0]

30 Mar 2020

All reviewers and editors have been addressed and the revised paper with a point by point response are attached.

---

## [Decision Letter · Decision Letter 1]

6 Apr 2020

Corneal Confocal Microscopy Identifies Greater Corneal Nerve Damage in Patients with a Recurrent Compared to First Ischemic Stroke

PONE-D-20-03573R1

Dear Dr. Malik,

We are pleased to inform you that your manuscript has been judged scientifically suitable for publication and will be formally accepted for publication once it complies with all outstanding technical requirements.

With kind regards,

Masaki Mogi

Academic Editor

PLOS ONE

Additional Editor Comments (optional):

No further comment.

Reviewers' comments:

Reviewer's Responses to Questions

**Comments to the Author**

1. If the authors have adequately addressed your comments raised in a previous round of review and you feel that this manuscript is now acceptable for publication, you may indicate that here to bypass the “Comments to the Author” section, enter your conflict of interest statement in the “Confidential to Editor” section, and submit your "Accept" recommendation.

Reviewer #1: All comments have been addressed

Reviewer #2: All comments have been addressed

2. Is the manuscript technically sound, and do the data support the conclusions?

Reviewer #1: Yes

Reviewer #2: (No Response)

3. Has the statistical analysis been performed appropriately and rigorously? 

Reviewer #1: Yes

Reviewer #2: (No Response)

4. Have the authors made all data underlying the findings in their manuscript fully available?

Reviewer #1: Yes

Reviewer #2: (No Response)

5. Is the manuscript presented in an intelligible fashion and written in standard English?

Reviewer #1: Yes

Reviewer #2: (No Response)

6. Review Comments to the Author

Reviewer #1: (No Response)

Reviewer #2: (No Response)

7. PLOS authors have the option to publish the peer review history of their article (what does this mean?). If published, this will include your full peer review and any attached files.

Reviewer #1: No

Reviewer #2: No

---

## [Editor Report · Acceptance letter]

10 Apr 2020

PONE-D-20-03573R1 

Corneal Confocal Microscopy Identifies Greater Corneal Nerve Damage in Patients with a Recurrent Compared to First Ischemic Stroke 

Dear Dr. Malik:

I am pleased to inform you that your manuscript has been deemed suitable for publication in PLOS ONE. Congratulations! Your manuscript is now with our production department. 

With kind regards,

on behalf of

Dr. Masaki Mogi 

Academic Editor

PLOS ONE